# Misleading Meta-Analyses during COVID-19 Pandemic: Examples of Methodological Biases in Evidence Synthesis

**DOI:** 10.3390/jcm11144084

**Published:** 2022-07-14

**Authors:** Erand Llanaj, Taulant Muka

**Affiliations:** 1ELKH-DE Public Health Research Group of the Hungarian Academy of Sciences, Department of Public Health and Epidemiology, Faculty of Medicine, University of Debrecen, Kassai út 26, 4028 Debrecen, Hungary; erand.llanaj@med.unideb.hu; 2Institute of Social and Preventive Medicine (ISPM), University of Bern, Mittelstrasse 43, 3012 Bern, Switzerland

**Keywords:** evidence synthesis, meta-analyses, COVID-19, public health

## Abstract

Not all evidence is equal. Evidence-based public health and medicine emanate from the principle that there is a hierarchy of evidence, with systematic reviews and meta-analyses (SRMAs) being at the top, as the highest level of evidence. Despite this, it is common in literature to find SRMAs with methodological issues that can distort the results and can thus have serious public health or clinical implications. During the Coronavirus Disease 2019 (COVID-19) pandemic, the importance of evidence and the way in which evidence was produced was stress tested and revealed a wide array of methodological biases that might have led to misleading conclusions and recommendations. We provide a critical examination of methodological biases in selected SRMAs on COVID-19, which have been widely used to guide or justify some pharmaceutical and nonpharmaceutical interventions with high public health and clinical significance, such as mask wearing, asymptomatic transmission, and ivermectin. Through these selected examples, we highlight the need to address biases related to the methodological quality and relevance of study designs and effect size computations and considerations for critical appraisal of available data in the evidence synthesis process for better quality evidence. Such considerations help researchers and decision makers avoid misleading conclusions, while encouraging the provision of the best policy recommendations for individual and public health.

## 1. Introduction

Evidence-based public health and medicine researchers have continuously exposed the fact that while many evidence synthesis efforts for widely used healthcare interventions show potential benefits, there is little high-quality evidence supporting their effectiveness and safety [1]. In the past years, there has been a massive explosion of unnecessary, misleading, and conflicted systematic reviews and meta-analyses (SRMAs) [2]. This phenomenon has persisted during the Coronavirus disease 2019 (COVID-19) pandemic. This has led to evidence synthesis efforts often serving mostly as easily produced publishable units or marketing tools, instead of promoting evidence-based public health. Although there are examples of evidence synthesis efforts that have supported best practices for healthcare during the pandemic [3], poor methodological quality and misleading SRMAs can be harmful given the major prestige and influence these types of studies have acquired. Even when rigorous SRMAs are too slow or difficult to conduct, the pandemic served as a reminder that it is still possible to recommend best practices and avoid methodological issues that are uncommon. Here, we present selected frequent methodological biases of SRMAs that had high visibility and were used to guide decisions on asymptomatic transmission, mask wearing, and ivermectin.

## 2. Asymptomatic Transmission in COVID-19

Information on asymptomatic transmission has high public health relevance as it dictates population-level measures, epidemiological practice, and routine outpatient care. In 2020, an SRMA of observational data showed the secondary attack rate was lower in contacts of people with asymptomatic infection than those with symptomatic infection (relative risk 0.35, 95% CI 0.10–1.27), suggesting that there is asymptomatic transmission, but at a lower rate compared to symptomatic transmission [4]. The article has been cited over 600 times and shared more than 1600 times on social media (e.g., Twitter).

In their analysis, the authors included five studies that conducted detailed contact investigations with sufficient data to estimate a secondary attack rate according to the symptom status of index cases. When examining the included studies, the authors appeared to have misclassified the preprint study by Chaw et al. [5], as reporting the secondary attack of asymptomatic COVID-19 patients, while this study actually provided the aggregate secondary attack for both presymptomatic and asymptomatic patients. Two months after the publication of the meta-analysis, the Chaw et al. paper was published in a peer-reviewed journal providing the secondary attack rate separately for asymptomatic patients (3 cases of 106 contacts) [6]. In addition, for studies having no events in a group, the authors added 0.5 to each cell in the 2 × 2 table. This arbitrary correction may lead to bias or even reverse the result of a meta-analysis, particularly when the sample sizes of the two groups are unbalanced. This is the case for two of the five studies included in the meta-analysis. For example, in the study by Cheng et al. [7], there were zero events in the asymptomatic group, and 22 events in the symptomatic group. However, the number of patients in the asymptomatic group was much smaller than in the symptomatic group, (91 vs. 2664); thus, adding 0.5 to arms that are unbalanced can distort the results, hence the authors’ conclusions. 

Similarly, in the study by Park et al. [8], describing a COVID-19 outbreak in a call center in South Korea, there were four patients in the asymptomatic group, with none of their contacts acquiring secondary infections, and 210 symptomatic patients with 34 secondary events, translating to a secondary attack rate of 16.2%. Adding 0.5 to such unbalanced arms can generate inaccurate conclusions (i.e., 0/4 vs. 0.5/4 is 0% vs. 12.5%). In such cases, the Freeman–Tukey double-arcsine variance-stabilizing transformation can address issues arising from the computation of the proportion in the presence of zero event counts [9]. When running the analysis (using *Metaprop* command in STATA 16.1 [10]) after taking this into account and applying inclusion and exclusion criteria provided by the authors, we find 0% asymptomatic transmission (95% CI: 0.00–0.00) and 5.3% (1.5–11.2%) symptomatic transmission (Figure 1). 

Additionally, adding results on asymptomatic transmission by Chaw et al., provided after peer review, generated similar conclusions (Figure 2). 

In another meta-analysis, Freeman-Tukey double-arcsine variance-stabilizing transformation was applied to address the issue arising from zero events, showing overall 1% (95% CI 0–2%) asymptomatic transmission [13]. However, when rerunning the analysis in STATA 16.1 based on data provided in the forest plot by the authors, the overall rate is 0.4% (95% CI 0–1.4%), suggesting on average 250 asymptomatic patients are needed to generate a single infection (Figure 3).

## 3. Wearing Facemasks and COVID-19

Facemasks are suggested to create a barrier against a high percentage of the viral particles released from a wearer’s mouth and nose [19], and high-quality medical masks may decrease particle emission during breathing [20]. A SRMA of mainly retrospective studies with poor methodological quality on mask wearing shows that they may reduce the risk of infection particularly in healthcare settings [21], but a SRMA of randomized controlled trials (RCTs) have not supported these findings [22]. The effectiveness of mask wearing in addition to adherence, quality, fitting, and other contextual factors is also subject to new emerging viral variants with higher viral loads [23]. During the pandemic, a SRMA showed that mask wearing was associated with a 53% reduction in COVID-19 transmission (95% CI; 0.29–0.75, I^2^ = 84%) [24]. In this analysis, the authors included data from six original studies. This article has been cited over 60 times, picked up by more than 130 media outlets, and shared more than 9000 times on social media (e.g., Twitter). 

There are several concerns regarding the analysis presented in this work. The first (major) concern was the inclusion in the meta-analysis of heterogeneous study designs, including an ecological study [25], one cross-sectional study [26], two case-control studies [27,28], one retrospective cohort study [29], and an RCT [30]. The merging of different study designs is methodologically problematic as the distinct study designs inherently carry dissimilar biases and answer research questions of different natures. For instance, while a prospective cohort study or RCT may be appropriate for a research question dealing with temporality, cross-sectional studies have very low relevance. In addition, ecological studies are generally used to generate a hypothesis [31].

Secondly, the authors appear to have missed other relevant studies, including two prospective cohort studies [32,33]. A prospective cohort study in Spain showed no significant association of risk of transmission with reported mask wearing by contacts (HR 1.55, 95%CI: 0.76–3.16) [32]. Similarly, a cohort study in Switzerland showed that constant mask wearing in public was not associated with infection risk in households (HR 0.94, 95% CI 0.43–2.09) [33]. Both these prospective studies were not included in the meta-analysis, and it is reasonable to assume that there are limitations associated with the search strategy of this SRMA, which may have led to omitting other relevant studies. Third, the authors considered the odds ratio (OR) equal to relative risk (RR) and combined these effect estimates together, which is a very common issue observed in meta-analyses. OR can be treated as RR under two circumstances according to Zhang et al. [34]: (i) if the OR value is between 0.5 and 2.5 or (ii) when there is a relatively low incidence of the observed outcome (<10%). If this is not the case, the OR should be converted to RR using specific formulas. The reported OR for the association of mask wearing and incidences of COVID-19 by Doung-Ngern et al. [28], Lio et al. [27], and Wang et al. [29] were all lower than 0.5 (0.23, 0.30, and 0.21, respectively), which could lead to biased estimates if the necessary adjustments are not performed. Instead, the overall conclusion of the paper could be that the use of a mask alone is not sufficient to provide protection against SARS-CoV-2 transmission, with higher-quality studies (e.g., prospective and clinical trials) showing in general a null effect. Similar issues have been observed also in another meta-analysis on this topic [35].

## 4. Ivermectin: An Example of How Critical Appraisal of Evidence Aids Good Science

Ivermectin, a horse dewormer and antiparasitic medication, has been widely promoted across the world for the treatment of COVID-19. After an in vitro experiment [36], a debate was fueled on the use of ivermectin as a therapy for COVID-19, with numerous subsequent studies challenging the initial in vitro study results.

A relatively small RCT compared an ivermectin dose of 12 mg/daily for 5 days (*n* = 22) or ivermectin 12 mg and doxycycline daily for 5 days (*n* = 23) against a placebo (*n* = 23) [37]. Ivermectin monotherapy resulted in reducing the time for viral clearance with a mean duration 9.7 days vs. 12.7 days for the placebo (*p* = 0.02), but ivermectin plus doxycycline did not show a comparable reduction (11.5 days). A preprint of a prospective controlled (nonrandomized) trial compared two to three doses of the drug in combination with 5 to 10 days of doxycycline (*n* = 70) with standard care (*n* = 70) [38]. Ivermectin/doxycycline therapy was associated with a reduced time to recovery of 10.6 days compared to 17.9 days for the placebo (*p* < 0.0001), but during the writing of this commentary, these results were not yet peer-reviewed or published.

A retrospective study of 280 patients hospitalized with confirmed COVID-19, reviewed the impact of ivermectin use in COVID-19 in four Florida hospitals [39]. The mortality was lower among patients who received ivermectin (*n* = 173, 15% mortality) compared to those receiving standard of care (*n* = 107, 25.2% mortality). After further adjustment, differences in mortality between the groups remained statistically significant (OR = 0.27, 95% CI: 0.09–0.80, *p* = 0.03). These findings, however, require randomized controlled trials for confirmation.

While meta-analyses on ivermectin and COVID-19 have shown a potential beneficial role of ivermectin, there are several serious methodological concerns observed in these meta-analyses, such as the inclusion of small and flawed RCTs [40] and positive results on ivermectin being driven mainly by methodologically low-quality studies [41]. However, critical evaluation from researchers has led to the retraction of the published meta-analyses. Nevertheless, this example has raised questions on the integrity of the evidence-generating process during COVID-19 [42]. A recent analysis has corroborated such concerns, as it showed that most healthcare interventions tested in *Cochrane Reviews* are not effective according to methodologically high-quality evidence [1]. Overall, these findings highlight the imperative of high-quality evidence synthesis and the importance of sharing data of individual studies, which would provide an opportunity to conduct meta-analyses of individual patient data (IPD).

## 5. Conclusions

Methodologically sound meta-analyses are important to not only improve our knowledge on a specific topic and understand gaps, but also to provide methodological rigor for scientific guidance for policy and decision-making. While pandemic situations require high-speed work in a context of tremendous professional pressure, it is important to preserve methodological rigor and ethical standards to ensure public health and clinical practice are informed by the best available evidence. Examples of methodological issues in meta-analyses related to COVID-19 are an indicator of common and repeated flaws in meta-analyses in general, generating misleading conclusions with serious health, economic, and social implications. While our work was not systematic, but rather focused on highly cited and influential meta-analysis in COVID-19, our paper highlights the importance of investigating systematically the extent to which meta-analyses published during the COVID-19 pandemic carry methodological issues that led to distorted conclusions. For high-quality and rigorous evidence synthesis, it is imperative for researchers to have a clear understanding of study methodological quality and designs, confounding and different types of biases, statistical models, and expertise of relevant meta-analytic methods for the specific topic under investigation [43]. What is more important, scientific inquiry should be driven by understanding the facts and seeking the truth, rather than using scientific methods to support preconceived perceptions or ‘eminence-based’ healthcare. Among others, using up-to-date methodologically sound guidelines for the conduct of meta-analysis, following PRISMA for reporting of systematic reviews [44], and applying GRADE (i.e., Grading of Recommendations, Assessment, Development, and Evaluations) [45] could help improve the quality of systematic reviews and meta-analysis and facilitate unbiased evidence-based recommendations.

## Figures and Tables

**Figure 1 jcm-11-04084-f001:**
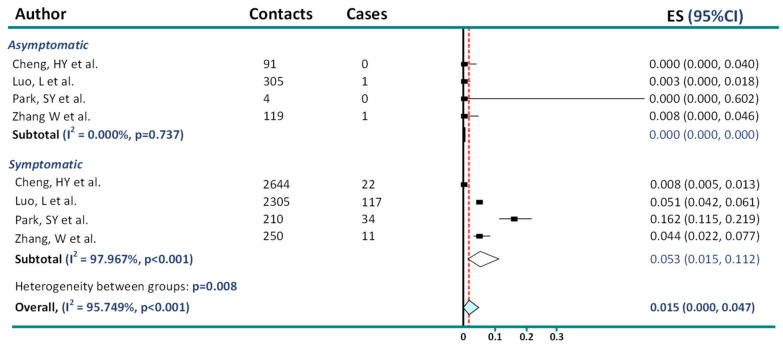
Reanalysis of data on asymptomatic and presymptomatic transmission [7,8,11,12].

**Figure 2 jcm-11-04084-f002:**
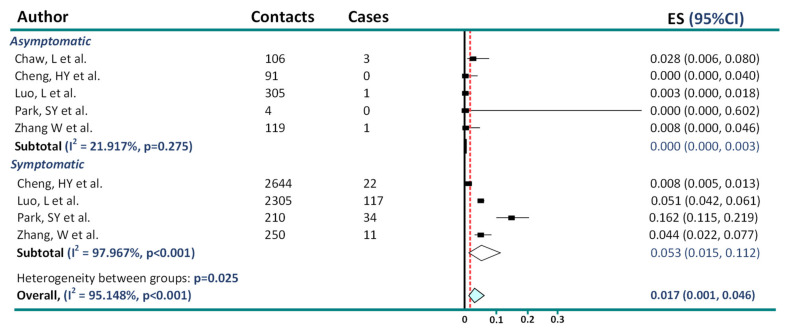
Reanalysis of data on asymptomatic and presymptomatic transmission adding Chaw et al. results [6,7,8,11,12].

**Figure 3 jcm-11-04084-f003:**
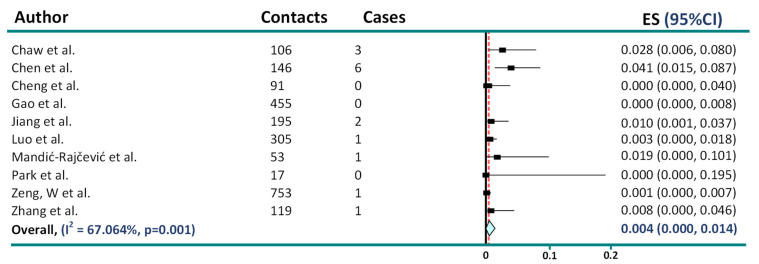
Reanalysis of secondary attack rates from truly asymptomatic and presymptomatic transmission [6,7,8,11,12,14,15,16,17,18].

## Data Availability

Not applicable.

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
