# Peer review of "Misleading Meta-Analyses during COVID-19 Pandemic: Examples of Methodological Biases in Evidence Synthesis"

_jcm, 2022, doi:10.3390/jcm11144084_

Round 1

Reviewer 1 Report

Dear Authors, 

A review should preferably include three sections dedicated to materials and methods, discussion, and conclusion. Therefore, the structure should be changed. With regards to the topic, unfortunately, I do not think that this paper may be of interest to the readers of the JCM. 

Sincerely

Author Response

Reviewer 1 Report / Responses

Comments and Suggestions for Authors

Dear Authors, 

A review should preferably include three sections dedicated to materials and methods, discussion, and conclusion. Therefore, the structure should be changed. With regards to the topic, unfortunately, I do not think that this paper may be of interest to the readers of the JCM.

REPLY. We agree with the Reviewer with regards to the fact that a review should have a more comprehensive outline and more sections. However, we did not intend to publish this as a review article. For this reason, we have requested the editor and the journal to change the type of the article from ‘Review’ to ‘Commentary’, for which our paper adheres to the requested formatting style. In addition, we have mentioned our manuscript is not a systematic appraisal of evidence, and future studies using systematic search and well-defined protocols are needed to quantify to what extent the meta-analyses published in Covid-19 suffer from methodological issues (lines 185-189 of the revised manuscript):

While our work was not systematic, but focused on highly cited and influential meta-analysis in COVID-19, our paper highlights the importance of investigating systematically to what extent meta-analyses published during COVID-19 pandemic carry methodological issues that lead to distorted conclusions.

Reviewer 2 Report

The authors proposed a review entitled “Misleading meta-analyses during COVID-19 pandemic: examples of methodological biases in evidence synthesis”. The aim to highlight the misleading publication is a very important issue and several efforts have been done to improve the knowledge of the evidence-based principles worldwide.

However, the authors should implement their manuscript according to the following points:

1)    Several methods to evaluate the body of evidence were developed during the years including some applied and recognized internationally such as The Grading of Recommendations Assessment, Development, and Evaluation (GRADE) method, however no related-references have been reported in the manuscript. Moreover, the PRISMA guidelines aiming to help authors improve the reporting of systematic reviews and meta-analyses should also be cited.

2)    List the type of methodological biases you considered and give a brief explanation (as a new paragraph if necessary). Moreover, in the abstract, please replace the sentence “certain type of biases” because it sound too generic with a more precise one.

3)    Explain more in detail the criteria for their inclusion of selected SRMA on COVID-19.

4)    I suggest to add for each research, the total number of meta-analysis available from which the authors selected the examples presented.

5)    At line, 138-139, references related to the studies challenging the initial in vitro study results and also of those related to the potential beneficial role of Ivermectin should be added.

6)    All the references should be checked according to the instructions for authors (https://www.mdpi.com/journal/jcm/instructions#references). For instance, in the reference 1, Year, Volume, page range should be added.

The authors proposed a review entitled “Misleading meta-analyses during COVID-19 pandemic: examples of methodological biases in evidence synthesis”. The aim to highlight the misleading publication is a very important issue and several efforts have been done to improve the knowledge of the evidence-based principles worldwide. The manuscript seems more a commentary than a review from my personal point of view, lacks of references to other methods and guidelines such as GRADE and PRISMA expecially for readers new to subject. However, I suggested to the author to include this information, a paragraph related to the methodological biases considered and the criteria for the inclusion of the selected example of SRMA on COVID-19 to ameliorate the manuscript.

Author Response

Reviewer 2 Report / Responses

Comments and Suggestions for Authors

The authors proposed a review entitled “Misleading meta-analyses during COVID-19 pandemic: examples of methodological biases in evidence synthesis”. The aim to highlight the misleading publication is a very important issue and several efforts have been done to improve the knowledge of the evidence-based principles worldwide.

REPLY. We thank the reviewer for highlighting the importance of the topic our paper covers.

However, the authors should implement their manuscript according to the following points:

1) Several methods to evaluate the body of evidence were developed during the years including some applied and recognized internationally such as The Grading of Recommendations Assessment, Development, and Evaluation (GRADE) method; however, no related-references have been reported in the manuscript. Moreover, the PRISMA guidelines aiming to help authors improve the reporting of systematic reviews and meta-analyses should also be cited.

DONE. We would like to thank the Reviewer for her/his time and valuable comments. We agree with the Reviewer with regards to the fact that a review should have a more comprehensive outline and tools such as GRADE and PRISMA should be utilized for a methodologically and reporting quality for a review article. We have now mentioned and referenced this briefly in the conclusion section of our manuscript (lines 185-188 of the revised manuscript):.

Among others, using up-to-date methodologically sound guidelines for the conduct of meta-analysis, following PRISMA for reporting of systematic reviews (36) and applying GRADE (Grading of Recommendations, Assessment, Development and Evaluations) (37) could help improve quality of systematic reviews and meta-analysis and facilitate unbiased evidence-based recommendations.”

2) List the type of methodological biases you considered and give a brief explanation (as a new paragraph if necessary). Moreover, in the abstract, please replace the sentence “certain type of biases” because it sound too generic with a more precise one.

DONE. This is an excellent suggestion. Based on the reviewers request we have modified the indicated sentence and divided in two parts as follow (lines 21-26):

 “Through these selected examples we highlight the need to address biases related to the methodological quality and relevance of study designs, effect size computations and considerations for critical appraisal of available data in the evidence synthesis process for better quality evidence. Such considerations help researchers and decision-makers avoid misleading conclusions, while encouraging provision of best policy recommendations for individual and public health.”

3) Explain more in detail the criteria for their inclusion of selected SRMA on COVID-19.

REPLY. Since we have requested a change in the type of article, we consider this unnecessary for the current type of article (i.e. commentary). The current articles were selected based on authors’ knowledge and the main topics debated during COVID-19 pandemic.

4) I suggest adding for each research, the total number of meta-analysis available from which the authors selected the examples presented.

REPLY. We did not perform a comprehensive search to quantify to what extend the examples we bring are observed in literature. Our paper is a commentary, and through examples of highly cited and influential meta-analysis in Covid-19, we would like to create awareness on the importance of methodological rigor in conducting meta-analysis. We, however, have mentioned this as limitation of our work and an important aspect to be investigated in the future (lines 185-189 of the revised manuscript):

While our work was not systematic, but focused on highly cited and influential meta-analysis in COVID-19, our paper highlights the importance of investigating systematically to what extent meta-analyses published during COVID-19 pandemic carry methodological issues that lead to distorted conclusions.

5) At line, 138-139, references related to the studies challenging the initial in vitro study results and of those related to the potential beneficial role of Ivermectin should be added.

REPLY. Based on the reviewers request we have included a brief and summarized description of additional studies showing benefits of Ivermectin (lines 146-162):

A relatively small RCT compared Ivermectin dose of 12 mg/daily for 5 days (n=22) or Ivermectin 12 mg and doxycycline daily for 5 days (n=23) against placebo (n=23) [29]. Ivermectin monotherapy resulted in reducing time for viral clearance with a mean duration 9.7 days vs. 12.7 days for placebo (p=0.02), but Ivermectin plus doxycycline did not show a comparable reduction (11.5 days). A preprint of a prospective controlled (non-randomized) trial compared 2 to 3 doses of the drug in combination with 5 to 10 days of doxycycline (n=70) with standard care (n=70) [30]. Ivermectin/doxycycline therapy was associated with reduced time to recovery of 10.6 days compared to 17.9 days for placebo (p<0.0001), but during the writing of this commentary, these results were not yet peer-reviewed or published.

A retrospective study of 280 patients hospitalized with confirmed COVID-19, reviewed the impact of Ivermectin use in COVID-19 in four Florida hospitals [31]. Mortality was lower among patients who received Ivermectin (n=173, 15% mortality) compared to those receiving standard of care (n=107, 25.2% mortality). After further adjustment, differences in mortality between groups remained statistically significant (OR=0.27, 95%CI: 0.09-0.80, p=0.03). These findings however require randomized controlled trials for confirmation.

6) All the references should be checked according to the instructions for authors (https://www.mdpi.com/journal/jcm/instructions#references). For instance, in the reference 1, Year, Volume, page range should be added.

REPLY. We have checked all references and formatted them according to the journal’s guidelines.

The authors proposed a review entitled “Misleading meta-analyses during COVID-19 pandemic: examples of methodological biases in evidence synthesis”. The aim to highlight the misleading publication is a very important issue and several efforts have been done to improve the knowledge of the evidence-based principles worldwide. The manuscript seems more a commentary than a review from my personal point of view, lacks of references to other methods and guidelines such as GRADE and PRISMA especially for readers new to subject. However, I suggested to the author to include this information, a paragraph related to the methodological biases considered and the criteria for the inclusion of the selected example of SRMA on COVID-19 to ameliorate the manuscript.

REPLY. We appreciate the reviewer sees the value of this work and we agree with reviewer that our manuscript falls more within a commentary. We have requested the editor and the journal to change the type of the article from ‘Review’ to ‘Commentary’. Moreover, we have now addressed the additional points the reviewer raised regarding use of guidelines in improving the quality of future meta-analysis, and importance of systematically appraising the evidence.

 (Lines 194-199 of the revised manuscript):

Among others, using up-to-date methodologically sound guidelines for the conduct of meta-analysis, following PRISMA for reporting of systematic reviews (36) and applying GRADE (Grading of Recommendations, Assessment, Development and Evaluations) (37) could help improve quality of systematic reviews and meta-analysis and facilitate unbiased evidence-based recommendations.”

(Lines 185-189 of the revised manuscript):

While our work was not systematic, but focused on highly cited and influential meta-analysis in COVID-19, our paper highlights the importance of investigating systematically to what extent meta-analyses published during COVID-19 pandemic carry methodological issues that lead to distorted conclusions.

Reviewer 3 Report

Thank for the opportunity to assess this paper. My specific comments are-

1. This paper is too short to consider as a review article. Mini review or brief/short report might be option.

2. Threre is no methodology for literature search strategy/keywords/search terms, these are are missing in the present form.

3. Background does not adequately drawn for this objective and need to discuss more evidences to justify the rationale of this paper.

4. Need a seperate section for discussion that will give evidence to validate the findings.

5. What are the limitations of the present study???

6. How the authors selected the papers for their analysis (forest plot curve)???

7. I do not find any justification to read as review article rather a short report. 

Author Response

Reviewer 3 Report / Responses

Comments and Suggestions for Authors

Thank for the opportunity to assess this paper. My specific comments are-

  1. This paper is too short to consider as a review article. Mini review or brief/short report might be option.
  2. There is no methodology for literature search strategy/keywords/search terms; these are missing in the present form.
  3. Background does not adequately drawn for this objective and need to discuss more evidences to justify the rationale of this paper.
  4. Need a separate section for discussion that will give evidence to validate the findings.
  5. What are the limitations of the present study???
  6. How the authors selected the papers for their analysis (forest plot curve)???
  7. I do not find any justification to read as review article rather a short report. 

REPLY. We would like to thank the Reviewer for her/his time and valuable comments. We agree with the Reviewer concerning the fact that a review should have a more comprehensive outline and tools such as GRADE and PRISMA should be utilized for a methodologically rigorous review article. We have mention this in lines 194-199 of the revised manuscript:

Among others, using up-to-date methodologically sound guidelines for the conduct of meta-analysis, following PRISMA for reporting of systematic reviews (36) and applying GRADE (Grading of Recommendations, Assessment, Development and Evaluations) (37) could help improve quality of systematic reviews and meta-analysis and facilitate unbiased evidence-based recommendations.

However, we did not intend to publish this as a review article. For this reason, we have requested the editor and the journal to change the type of the article from ‘Review’ to ‘Perspective’. We believe that in this format fits the style of paper and in this way; our work may be more interesting for the readers of the Journal of Clinical Medicine.

Round 2

Reviewer 3 Report

No comments